# Cortactin Contributes to Activity-Dependent Modulation of Spine Actin Dynamics and Spatial Memory Formation

**DOI:** 10.3390/cells10071835

**Published:** 2021-07-20

**Authors:** Jonas Cornelius, Klemens Rottner, Martin Korte, Kristin Michaelsen-Preusse

**Affiliations:** 1Division of Cellular Neurobiology, Zoological Institute, TU Braunschweig, 38106 Braunschweig, Germany; j.feuge@tu-braunschweig.de (J.C.); m.korte@tu-braunschweig.de (M.K.); 2Research Group Molecular Cell Biology, Helmholtz Centre for Infection Research, 38124 Braunschweig, Germany; k.rottner@tu-braunschweig.de; 3Division of Molecular Cell Biology, Zoological Institute, TU Braunschweig, 38106 Braunschweig, Germany; 4Research Group Neuroinflammation and Neurodegeneration, Helmholtz Centre for Infection Research, 38124 Braunschweig, Germany

**Keywords:** cortactin, learning, memory formation, actin, hippocampus, cytoskeleton, actin-binding protein, structural plasticity, functional plasticity

## Abstract

Postsynaptic structures on excitatory neurons, dendritic spines, are actin-rich. It is well known that actin-binding proteins regulate actin dynamics and by this means orchestrate structural plasticity during the development of the brain, as well as synaptic plasticity mediating learning and memory processes. The actin-binding protein cortactin is localized to pre- and postsynaptic structures and translocates in a stimulus-dependent manner between spines and the dendritic compartment, thereby indicating a crucial role for synaptic plasticity and neuronal function. While it is known that cortactin directly binds F-actin, the Arp2/3 complex important for actin nucleation and branching as well as other factors involved in synaptic plasticity processes, its precise role in modulating actin remodeling in neurons needs to be deciphered. In this study, we characterized the general neuronal function of cortactin in knockout mice. Interestingly, we found that the loss of cortactin leads to deficits in hippocampus-dependent spatial memory formation. This impairment is correlated with a prominent dysregulation of functional and structural plasticity. Additional evidence shows impaired long-term potentiation in cortactin knockout mice together with a complete absence of structural spine plasticity. These phenotypes might at least in part be explained by alterations in the activity-dependent modulation of synaptic actin in cortactin-deficient neurons.

## 1. Introduction

Activity-dependent plasticity is associated with functional as well as structural changes at synapses and plays a vital role both during development and for cognitive functions as, for instance, processes of learning and memory formation [1,2,3]. The main underlying force needed for structural changes at synapses as well as for alterations in the composition of surface receptors in response to external stimuli is generated by the actin cytoskeleton, particularly via actin-binding proteins (ABPs) [4,5]. The ABP cortactin (Cttn) is of special relevance in neurons, as it has been shown to be important both for pre- and postsynaptic structures [6,7,8,9,10,11]. Cttn is predominantly enriched in dendritic spines, which contain the majority of excitatory synapses in the cortex, and Cttn might play a major role in spine formation and morphology [12]. Notably, loss of Cttn reduces the dendritic spine number [7]. Cttn possesses an N-terminal acidic domain that interacts with the F-actin branching Arp2/3 complex, an actin-filament-binding region and a Src homology 3 (SH3) domain, which allows for interaction with other effectors of the actin cytoskeleton or scaffolding proteins [13]. Interestingly, Cttn has the unique property of being able to bind filamentous actin as well as the Arp2/3 complex simultaneously and, in line with this, the subcellular localization of Cttn depends on interactions with both [14]. Notably, Cttn has a 15-fold higher affinity for ATP-bound actin than ADP-bound actin [15], suggesting that it prefers binding to newly generated actin networks. It is believed that Cttn contributes to both the activation and, in particular, the stabilization of Arp2/3 in branch junctions, yet the exact molecular mechanisms of its Arp2/3-dependent functions remain elusive. Cttn was described to compete with Arp2/3 regulators of the WASP family for binding to at least one of the two distinct class I nucleation promoting factor (NPF) binding sites on Arp2/3 [16], and while it weakly activates Arp2/3 on its own, it is believed to potently activate Arp2/3 through synergistic action together with other NPFs such as N-WASP [17,18]. Importantly, however, Arp2/3-dependent actin networks can form normally even in the absence of Cttn [19,20], suggesting that Cttn might serve a tuning function rather than being obligatory for Arp2/3 activation in vivo. In addition, Cttn might also contribute to F-actin dynamics through Arp2/3-independent mechanisms as it has also been shown to mediate GTPase activation through a yet unknown mechanism [19].

Intriguingly, Cttn has been found to be crucial for several neuron-specific functions such as axon guiding [21], synaptogenesis [7,12] or growth cone formation [22]; however, its importance for activity-dependent modifications of synapse structure and function remains unknown. In dendritic spines, Cttn interacts with scaffolding proteins of the Shank family at the postsynaptic density [23] and is thereby indirectly connected to ionotropic and metabotropic glutamate receptors. In addition, it was shown to directly interact with voltage-gated K^+^ channels [24] as well as the Ca^2+^-sensor caldendrin [25], indicating its potential to dynamically regulate synaptic efficacy and to couple Ca^2+^ signaling to structural modulations of synaptic actin. Finally, further emphasizing an important role for synaptic plasticity processes at dendritic spines, Cttn distribution is mediated by neuronal activity as NMDA-receptor activation induces a Src family-dependent Cttn phosphorylation and translocation from dendritic spines to dendrites [8]. In addition, BDNF triggers ERK-dependent phosphorylation of Cttn, inducing its redistribution from dendrites into dendritic spines [8]. In summary, evidence points towards a potentially relevant role of Cttn in activity-dependent modifications of synaptic function and structure, yet its precise functions in neurons and especially for cognitive processes are still unclear.

Thus, in this work, we analyzed in so far neurologically uncharacterized *Cttn* knockout mice [26] the consequence of Cttn removal for spatial memory formation, functional and structural plasticity processes as well as activity-dependent modulation of spine actin dynamics. Interestingly, we show here that Cttn-deficient mice display deficits in memory recall together with an impairment in long-term potentiation and a complete lack of structural spine plasticity. Finally, our data reveal impairments in the activity-dependent modulation of synaptic actin as a potential underlying mechanism for the observed cognitive deficits.

## 2. Materials and Methods

**Mice.** All experiments were authorized by the LAVES (Oldenburg, Germany, Az. §4 (02.05) TSchB TU BS, 33.19-42502-04-11/0679, 33.19-42502-04-20/3498) and the animal welfare representative of the TU Braunschweig. Mice carrying deleted alleles of *Cttn* upon Cre-recombinase-mediated excision of exon 7, as described in [26], were backcrossed to C57BL/6 background (>10x) and then used for experiments as follows. All mice were kept in standard conditioned cages and were exposed to a 12 h dark/light cycle. In addition, mice were specifically bred for preplanned experiments and care was taken to ensure that the handling of mice was identical across all experiments. For all experiments, the experimenter was blinded to genotype and mice were chosen from cages in a random order.

**Genotyping.** For genotyping, genomic DNA was obtained from ear biopsies and genotyping of Cttn-deficient mice was performed as described. The following primer pairs (called 2 and 5) were used: Fwd2 5′-CCT GGA ATA AGT CAG CCA AGC–3′ and Rev2 5′-ATG GCC CTA GAG GTC AAG C–3′ for detection of the WT allele, and Fwd5 5′-AGG GTC TGA CCA TCA TGT CC–3′ and Rev5 5′-GTG CTG TTC ATC CAC AAT GC–3′ for detection of the deleted allele.

**Behavioral analysis in the Open Field arena.** All test trials were done in an arena 40 × 40 × 40 cm wide (custom build). The center zone was specified as the 10 × 10 cm central region, the middle zone was specified as the 30 × 30 cm central region, and the remaining outer 10 cm were defined as the border region. Mice were placed along one side of the field and had 5 min of exploration time. Between individual sessions, the whole apparatus was extensively cleaned to remove remaining odor traces. All trials were recorded using a digital video camera and Any-maze tracking software (Stoelting Europe, Dublin, Ireland, Version 6.34).

**Spatial learning in the Morris Water Maze.** All test trials were done in a 160 cm wide tank (TSE Systems GmbH, Bad Homburg, Germany)with water temperatures of 19–21 °C and all mice underwent pre-training for 3 consecutive days (2 trials per day) with a visible platform to get accustomed to the pool, the experimenter as well as the training situation. After a 48 h break, all mice were then trained for 8 consecutive days in the hidden platform version of the task, with 4 trials per mouse each day. Specifically, mice were placed in one of four different starting locations in the pool and had 60 s time to locate the submerged platform. If an animal failed to locate the platform, it was guided to it and remained there for 20 s before being put back into its cage. Reference memory tests were done on training days 3, 6 and 9, on which one probe trial was performed with the platform removed and animals having to search for 45 s. All trials were recorded and analyzed individually utilizing a digital video camera as well as the Any-maze tracking software (Stoelting Europe, Dublin, Ireland, Version 6.34).

**Acute hippocampal slice preparation.** Brains were removed and quickly transferred into cold 95% O_2_ and 5% CO_2_ saturated artificial cerebrospinal fluid (aCSF-125 mM NaCl, 2.5 mM KCl, 2 mM MgCl_2_, 2 mM CaCl_2_, 1.25 mM NaH_2_PO_4_, 26 mM NaHCO_3_, 25 mM glucose (all chemicals: Applichem, Darmstadt, Germany)). Subsequently, the hippocampi were isolated and immediately sliced into 400-µm-thick slices using a vibratome (VT1200S, Leica Microsystems, Wetzlar, Germany). Finally, all slices were stored in 95% O_2_ and 5% CO_2_ saturated aCSF for 90 min prior to further analysis.

**Primary embryonic hippocampal cultures** were prepared at embryonic day 18 as described previously [27]. In total, 70,000 cells were plated on 13 mm poly-l-lysine-coated coverslips and were maintained at 37 °C, 5% CO_2_ (vol/vol) and 99% humidity in Neurobasal medium (Gibco (ThermoFischer Scientific), Waltham, MA, USA), which was additionally supplemented with 10% N2 (vol/vol), 2% B27 (vol/vol) and 0.5 mM Glutamax (all: Gibco (ThermoFischer Scientific), Waltham, MA, USA). All cultures were then used for experiments either after 14 or 21 days in culture.

**Transfection of cultured hippocampal neurons.** Primary embryonic hippocampal cultures were transfected after 14 or 21 days in vitro using Lipofectamine 2000 (Invitrogen (ThermoFischer Scientific), Waltham, MA, USA) according to the manufacturer’s instructions. We used 2 µg of Lipofectamine and 1 µg plasmid-DNA of one of the following plasmids per well (24-well plate): EGFP-tagged β-actin for fluorescence recovery after photobleaching experiments or farnesylated-EGFP (both Clontech (Takara Bio Inc.), Kusatsu, Shiga, Japan) to visualize individual neurons. Cultures were then used for experiments 48 h after transfection.

**cLTP Induction in primary embryonic hippocampal cultures.** All cells were incubated at room temperature for 20 min in 1× Hanks Balanced Salt Solution (HBSS, 37 °C pre-heated; Gibco (ThermoFischer Scientific), Waltham, MA, USA) and afterwards stimulated for 10 min using Mg^2+^-free HBSS containing 200 µM glycine (Applichem, Darmstadt, Germany) and 3 µM strychnine (Sigma-Aldrich (Merck KGaA), Darmstadt, Germany). Immediately afterwards, the Mg^2+^-free HBSS was replaced with normal HBSS and cells were incubated for up to additional 50 min.

**Electrophysiology.** Acute hippocampal slices were submerged into a recording chamber and all recordings performed at 32 °C. **Excitatory postsynaptic potentials** (EPSPs) were recorded in the *stratum radiatum* of the hippocampal CA1 region utilizing a glass micropipette filled with 3M NaCl (Applichem, Darmstadt, Germany) at a depth of approximately 150 to 200 µm (resistance: 3–15 mΩ). To elicit an EPSP with a slope of ~40% of the maximum slope, Schaffer collaterals were stimulated at a frequency of 0.1 Hz. **Input–output recordings** were performed by application of a predefined current of 25 to 250 µA (in steps of 25 µA) while **paired-pulse facilitation** was done by application of two stimuli spaced by varying interstimulus intervals (10, 20, 40, 80, 160 ms). After 20 min of baseline recording, **long-term potentiation** (LTP) was induced by theta-burst stimulation (TBS). Three bursts at 0.1 Hz were used, of which each burst consisted of 4 pulses at 100 Hz repeated 10 times in 200 ms intervals (5 Hz). Presynaptic properties as well as basal synaptic transmission were analyzed via paired-pulse facilitation and input–output measurements. All data were collected and analyzed with LABVIEW software (National Instruments, Austin, Texas, USA, Version 4.751). The slope of EPSPs after stimulation of Schaffer collaterals was measured over time, normalized to baseline recordings and finally plotted as average ± SEM.

**Imaging and image analysis.** Imaging was done with a confocal laser scanning microscope (Olympus Fluoview1000, Olympus Deutschland GmbH, Hamburg, Germany) equipped with 20× (0.5 NA), 40× (1.3 NA) and 60× (1.0 NA) objectives, and resulting images were analyzed using ImageJ (National Institutes of Health, Bethesda, MD, USA, Version 1.52a). For the analysis of dendritic spine properties, we quantified average dendritic spine head diameters as well as total number of dendritic spines (spine density: spines per µm dendrite) on given dendritic segments. For the analysis of dendritic spine motility, we imaged the same dendritic segments every 5 min for 20 min followed by analysis of absolute as well as mean changes in head diameters as well as lengths of individual dendritic spines.

**Fluorescence recovery after photobleaching (FRAP).** FRAP experiments were performed using primary embryonic hippocampal cultures (either after 14 or 21 days in culture), in which EGFP-tagged β-actin fluorescence was bleached in individual dendritic spines using the 405 nm laser line at a power of 2.3 to 3 mW (approx. 30%) for 150 ms. Simultaneously, EGFP emission at 488 nm was recorded using a SIM scanner (Olympus) and a time-lapse series was recorded for app. 180 s following photobleaching with a time interval of 3 s (65 images in total: 5 images pre-bleach, 60 post-bleach). All images were analyzed using ImageJ software (National Institutes of Health, Bethesda, MD, USA, Version 1.52a). In brief, mean fluorescence intensity values for each time point were normalized to average intensities derived from 5 pre-bleaching images for each spine individually, and plotted against time. Subsequently, utilizing the GraphPad Prism software (GraphPad Software, San Diego, CA, USA, Version 5.01), nonlinear curve fitting of net recovery curves after photobleaching was performed using the following equation: Y = Y0 + (Plateau − Y0) × (1 − exp(−K × x)), where Y0 equals Y value at time point zero after bleaching. The plateau corresponds to the Y value at infinite times, and is expressed as the fraction of fluorescence before bleaching. This value was used for determining dynamic and stable actin pools (the stable pool is the fraction of fluorescence that does not recover within the imaging period of 3 min calculated as 1 − (dynamic F-actin)), K is the rate constant, and τ is the time constant, expressed in seconds; it is computed as the reciprocal of K. From this equation, the actin turnover rate was calculated as the time point at which 50% of pre-bleaching fluorescence levels were reached.

**Data presentation and statistical analysis.** If not specifically stated otherwise, data are depicted means ± SEMs and an α level of *p* < 0.05 was used as criterion to reject the null hypothesis. Normal distribution was tested utilizing the Kolmogorov-Smirnov Test. Comparisons between two different groups were done using unpaired Student’s *T*-tests while comparisons of two or more groups with varying treatments were analyzed by Two-Way ANOVA (parametric data). For non-parametric data, a Kruskal-Wallis Test followed by a Dunn’s Test was performed. A complete list of statistical tests and values can be found in Appendix A.

## 3. Results

### 3.1. Cttn-Deficient Mice Show Deficits in Hippocampus-Dependent Spatial Memory Formation

To shed light on the physiological role of cortactin (Cttn) in neurons, we analyzed the behavior and brain function of a previously generated *Cttn*-knockout (KO) mouse line in the C57BL/6 background [26], which hitherto had not been analyzed with respect to neuronal phenotypes. First, to assay general locomotor activity, we analyzed the behavior of *Cttn*-KO mice and C57BL/6 wildtype (WT) littermates in the Open Field Test [28] (Figure 1A). In comparison to WT animals, the behavior of Cttn-deficient mice was similar in regard to the relative time in the border versus the central region of the arena (Figure 1A) as well as the total distance traveled (WT 2.606 ± 0.173 m; KO 2.982 ± 0.164 m), suggesting that *Cttn*-KO mice show normal exploration behavior and do not experience increased levels of anxiety. In addition, no locomotor deficits were observable in any of the knockout mice over the course of the experiment. These observations allowed us to further test these animals in the Morris Water Maze task [29], a test that assesses cognitive function and hippocampus-dependent spatial memory formation (Figure 1B–F). Again, locomotor abilities were normal in Cttn-deficient animals as, over the whole training period, the average (0.198 ± 0.004 m/s) as well as the maximum swim speed (0.304 ± 0.004 m/s) were nearly identical in comparison to wildtype mice (average 0.196 ± 0.004 m/s; maximum 0.307 ± 0.003 m/s). During the 8 days of training, mice of both genotypes showed a progressive reduction in escape latency, indicating memory formation (Figure 1B). However, Cttn-deficient animals displayed a higher latency compared to WT mice throughout the training (Figure 1B). In addition, *Cttn*-knockout animals were significantly less accurate in spatial reference memory tests performed always before the training on day 3 and 6 (Figure 1C–E). Only on the last reference memory test performed 24 h after the last training session on day 9, both genotypes performed equally well (Figure 1F).

In summary, these results suggest that the knockout of *Cttn* leads to a moderate deficit in hippocampus-dependent spatial memory formation.

### 3.2. Cttn-Deficient Mice Show Impaired Synaptic Plasticity at The Hippocampal Schaffer Collateral Pathway

Impairments in spatial memory formation and recall pointed towards deficits in synaptic plasticity as an underlying mechanism involved in the storage of information in neuronal networks. Therefore, we studied general neuronal functionality and functional plasticity in greater detail. Analysis of excitatory postsynaptic potentials (EPSP) (Figure 2A) and input–output measurements (Fiber Volley) (Figure 2B) revealed no alterations in basal synaptic transmission in *Cttn*-KO mice and the same was true for short-term presynaptic plasticity as evaluated via paired-pulse facilitation (Figure 2C). However, when we induced long-term potentiation (LTP) after 20 min of baseline recording at the Schaffer collateral pathway connecting the CA3 and CA1 region via theta-burst stimulation in acute hippocampal slices of *Cttn*-KO as well as WT mice (Figure 2D), LTP was significantly impaired in the absence of Cttn as compared to controls. Whereas WT mice showed a potentiation of 134.75 ± 0.33% in the last 5 min of the recording, *Cttn*-KO mice displayed a statistically significantly lower potentiation of 125.04 ± 0.79% (Figure 2D).

Taken together, our results suggest that as a result of the loss of Cttn, functional plasticity mechanisms are dysregulated in the hippocampal neurons of Cttn-deficient mice.

### 3.3. Basal Actin Dynamics and Dendritic Spine Motility Are Unaltered in Cttn-Deficient Hippocampal Neurons

As our electrophysiological experiments showed an impairment in long-term potentiation in the absence of Cttn, we next aimed at analyzing structural plasticity processes and especially the dynamics of actin in *Cttn*-KO neurons or more specifically at dendritic spines. As a first step, we were interested in testing whether spine actin dynamics would be altered per se in the absence of Cttn. Therefore, primary embryonic hippocampal cultures were transfected to express EGFP-tagged ß-actin. This allowed us to perform fluorescence recovery after photobleaching experiments at individual dendritic spines, assessing actin filament turnover times as well as stable versus dynamic pools of spine actin. The experiments were performed using developing (14 days in vitro) as well as mature (21 days in vitro) hippocampal neurons derived from WT versus *Cttn*-KO littermate mice (Figure 3A). We found that basal actin dynamics were completely unaltered both at 14 days as well as at 21 days in culture comparing both genotypes (Figure 3A,B). Interestingly, however, in WT as well as Cttn-deficient hippocampal neurons, F-actin turnover times (the time point where 50% of the original fluorescence recovered) differed depending on time in culture (Figure 3B), since actin turnover in 14-day-old cultures was generally higher than in 21-day old-cultures, suggesting that actin dynamics do indeed change over the time course of maturation. In contrast to this, the relative sizes of stable versus dynamic actin fractions (the amount of actin fluorescence which recovered (dynamic) or did not recover (stable) in the time period of 180 s after the photobleach; analyzed over 3 min) were similar in both 14- and 21-day-old cultures.

In the next set of experiments, we additionally analyzed dendritic spine motility, which is dependent on modulations of the spine actin cytoskeleton [4], and thus constitutes a more indirect readout of potential alterations in actin dynamics upon Cttn loss. For this, we quantified absolute as well as mean changes in dendritic spine head diameter and dendritic spine length in time-lapse recordings from a time course of 20 min in cultured, EGFP-expressing hippocampal neurons of both genotypes (Figure 3C–E). Again, we compared developing spines of neurons that were 14 days in culture with mature spines of cells cultivated for 21 days. In line with our observation that baseline spine actin dynamics were not affected by the loss of Cttn, dendritic spine motility was also unchanged in *Cttn*-KO cells compared to WT neurons (Figure 3D,E). Neither absolute/mean changes in dendritic spine head diameter (Figure 3D) nor absolute/mean changes in dendritic spine length (Figure 3E) were altered in hippocampal neurons derived from Cttn-deficient mice compared to WT.

### 3.4. Structural Spine Plasticity Is Impaired in the Absence of Cortactin

Since baseline synaptic properties as well as basal actin dynamics were unaltered in the absence of Cttn, whereas spatial learning as well as functional plasticity were significantly impaired, we hypothesized that Cttn might be involved in specifically mediating activity- or experience-dependent synaptic alterations. As a next step, therefore, we investigated whether structural spine plasticity would be affected by the loss of Cttn. Again, in line with intact basal synaptic transmission in *Cttn*-KO animals, we found that spine densities (Figure 4A) and spine morphologies (spine head diameter, Figure 4B) per se were not altered in the absence of Cttn. We then chemically induced NMDAR-dependent long-term potentiation in cultures derived from WT or *Cttn*-KO mice (via glycine and strychnine) and analyzed average dendritic spine head diameters 60 min after stimulation. While, in WT neurons, a significant increase in spine head size upon cLTP induction was observed (ctrl 0.562 ± 0.013 µm to cLTP 0.690 ± 0.017 µm), this was completely lacking in Cttn-deficient hippocampal neurons (ctrl 0.587 ± 0.014 µm to cLTP 0.555 ± 0.017 µm) (Figure 4B). Thus, Cttn deficiency completely abrogated activity-dependent structural plasticity.

Next, we speculated that this impairment in spine structural plasticity might indeed result, at least in part, from a dysregulation of activity-dependent spine actin dynamics in Cttn-deficient neurons. Thus, we compared the results of FRAP experiments performed under baseline conditions to fluorescence recovery after photobleaching of EGFP-tagged ß-actin at individual dendritic spines 15 min after chemical induction of long-term potentiation (Figure 4C–E). In WT neurons, the turnover time of F-actin slightly increased, while the dynamic fraction of spine actin moderately but statistically significantly decreased in this time period (Figure 4C,E). This is in line with a dynamic modulation of the spine actin cytoskeleton that typically occurs in this early phase of long-term potentiation. Intriguingly, actin dynamics in Cttn-deficient hippocampal neurons were altered differently following cLTP induction, as we observed a significant increase in F-actin turnover times while the ratio of dynamic and stable actin fractions did not change, as opposed to what was observed in WT spines (Figure 4D,E).

In summary, our results reveal that in line with the observed deficits in spatial learning and functional plasticity, structural spine plasticity as well as the underlying regulations of the dendritic spine actin cytoskeleton are dysregulated in Cttn-deficient hippocampal neurons.

## 4. Discussion

Extensive evidence points towards a close association of memory formation, functional synaptic plasticity and structural changes in dendritic spines [30,31]. While it is well established that these structural adaptations are dependent on regulations of synaptic actin [32], also functional changes at synapses involve a reorganization of the actin cytoskeleton [33]. In line with this, long-term potentiation (LTP) can be efficiently blocked through interference with actin polymerization [5,34,35]. However, although the intimate connection between F-actin, synapse structure and synapse function had been described already decades ago, the cellular mechanisms and the regulations of actin-modulating proteins that convert neuronal activity into morphological change have mostly remained elusive. One of the actin regulators that harbors the potential to mediate both structural and functional changes at dendritic spines is cortactin (Cttn) [36], an actin-binding protein whose function in neurons is still largely unknown. At the synapse, Cttn has been shown to directly interact with scaffolding Shank family proteins of the postsynaptic density (PSD) [23], with voltage-gated K^+^ channels [24] as well as with the Ca^2+^ sensor caldendrin [25], and, thus, might be able to efficiently link Ca^2+^-signaling to dynamic modifications of synaptic actin and synaptic efficacy during processes of synaptic plasticity. In addition, Cttn has been implicated as an activator or at least a stabilizer of actin filament branches formed by the Arp2/3 complex [13,17,37,38]. In this respect, Cttn can be considered a relevant regulator of virtually all actin structures, at least those harboring the Arp2/3 complex, not only in neurons.

In the present study, we took advantage of a previously described *Cttn*-KO mouse line [26], and showed that Cttn indeed does play a crucial role in functional as well as structural synaptic plasticity, and eventually even complex cognitive processes such as learning and memory formation. Our data suggest that, as a result of the loss of Cttn, hippocampal neurons show a dysregulation in the activity-dependent modulation of actin dynamics in dendritic spines. Importantly, this deficit was specific for synaptic plasticity as basal synaptic properties, spine density as well as actin dynamics under baseline conditions were not affected by the loss of Cttn. At first glance, our results appeared counter-intuitive since dendritic spine structure as well as maintenance are known to be Arp2/3 complex-dependent, with a gradual loss of spines in conditional ArpC3 KO mice [39]. Moreover, one of the main functions initially described for this protein had been the modulation of Arp2/3 complex activation [38]. However, in line with our present data, previous studies already suggested that Cttn function might indeed be dependent on the context and its molecular regulation, not only in neurons. In line with this, Cttn removal did not reduce lamellipodial Arp2/3 complex activation and stabilization in fibroblasts [19] or Arp2/3-dependent podosome formation in megakaryocytes [20], suggesting that, in vivo, Cttn may tune Arp2/3-dependent actin structures in a context-dependent manner instead of operating in Arp2/3 complex activation. Our data suggest that in neurons, Cttn function is predominantly regulated in the context of synaptic activity. Supporting this hypothesis, we provide evidence that the loss of Cttn leads to a complete lack of structural plasticity in hippocampal neurons in vitro as well as to reduced long-term potentiation in acute hippocampal slices at the Schaffer collateral to CA1 pathway. Most importantly, we can show that these dysfunctional synaptic plasticity mechanisms may indeed manifest as deficits in hippocampus-dependent spatial memory formation in Cttn-deficient mice. The data provided here thus point at a direct correlation between Cttn activity and cognitive function.

Interference with Arp2/3 complex activity in neurons leads to a profound loss of dendritic spines and abnormal dendritic spine morphology [39], which further supports the hypothesis that the phenotypes observed in Cttn-deficient mice likely derive from a context-dependent modulation of actin dynamics rather than defects in Arp2/3 complex activity. Phenotypes comparable to interference with Arp2/3 complex function [39] were indeed observed upon deletion of the C-terminal VCA region of N-WASP (which binds and activates the Arp2/3 complex) [40] or after loss of other activators of the Arp2/3 complex such as WAVE-1 [41,42]. Both N-WASP and WAVE family members belong to the class II NPFs such as Cttn [43], the latter of which might indeed have different functions in actin dynamics in vivo [13]. Consistently, the aforementioned phenotypes could not be observed in *Cttn*-knockout animals. Interestingly, the specific increase in F-actin turnover upon cLTP induction in the spines of Cttn-deficient hippocampal neurons was reminiscent of increased actin filament and Arp2/3 complex turnover in cortactin-depleted lamellipodia [19] or increased Vaccinia tail velocities upon Cttn RNAi [44]. Although it is tempting to speculate that all these observations have common causes, future experiments will have to solidify this view.

Notwithstanding this, and as opposed to the dysregulated Arp2/3 complex, attractive hypotheses potentially explaining the phenotypes of Cttn-deficient mice described here include a dysregulation of the Rho subfamily of small GTPases, such as Cdc42 and Rac1, which were shown to be significantly less activated as a result of Cttn loss in fibroblasts [19]. In line with this, mice deficient in either Rac1 or Cdc42 share comparable phenotypes with *Cttn*-knockout animals. The loss of Rac1 activity, either via knockout [45,46] or through pharmacological inactivation [47], has been shown to impair long-term plasticity in the hippocampus, and furthermore, Rac1-deficient mice show impairments in hippocampus-dependent spatial memory formation [45]. Similarly, structural and functional plasticity in hippocampal neurons was abolished in forebrain-specific, conditional Cdc42 knockout animals, which was accompanied by deficits in remote memory recall, whereas working memory, anxiety levels and locomotor activities were unaltered [48]. Given the similarities to the molecular and behavioral phenotypes of *Cttn*-knockout mice described here, future studies will be needed to unravel details about a potentially exciting new role of Cttn as a regulator of RhoGTPase activity in the nervous system. In this respect, it is important to emphasize that targeted activation of Rho GTPases has been shown to improve learning and memory [49]. Notably, a detailed understanding of all these phenomena might be particularly relevant for a better understanding of certain psychiatric disorders, e.g., in schizophrenic patients, in which expression levels of both Cttn and Cdc42 were found to be significantly reduced [50,51].

Taken together, our experiments in *Cttn*-knockout mice show a key role of Cttn in synaptic plasticity, activity-dependent structural modulation and learning behavior. The loss of Cttn inhibits the activity-dependent dynamics of synaptic actin and we can correlate this to a complete lack of structural plasticity and decreased LTP in hippocampal neurons. Thus, our data suggest a direct link between Cttn signaling, proper neuronal function and, ultimately, cognitive abilities.

## Figures and Tables

**Figure 1 cells-10-01835-f001:**
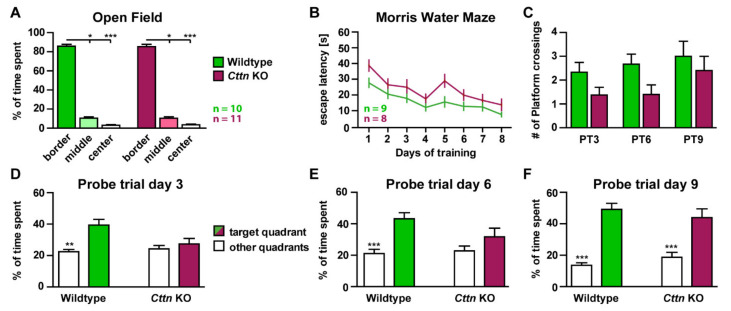
**Behavioral analysis of Cortactin-deficient mice.** (**A**) Wildtype and Cttn-deficient mice trained in the Open Field test [28]. Depicted is the percentage of time spent in the center, middle and border region of the test arena. n = number of animals. (**B**) Escape latency of male WT and Cttn-deficient animals trained for 8 days in the hidden platform version of the Morris Water Maze [29]. n = number of animals. (**C**) Amount of platform crossings in reference memory tests (‘Probe trials’, PT) performed on training day 3, 6 and 9. (**D**–**F**) Percentage of time spent in the target quadrant (the quadrant where the hidden platform was originally located) and the other quadrants in the reference memory test on (**D**) training day 3, (**E**) day 6 and (**F**) day 9. **Data information:** Statistic tests used: For normal distribution of data: Kolmogorov–Smirnov Test, Parametric Tests: Two-Way ANOVA and Sidak’s Multiple Comparisons, Non-Parametric Tests: Kruskal-Wallis Test and Dunn’s test; data depicted as means ± SEM, detailed information on *p* values as well as statistical tests is depicted in Appendix A; significances are indicated by * *p* value < 0.05, ** *p* value < 0.01 and *** *p* value < 0.001.

**Figure 2 cells-10-01835-f002:**
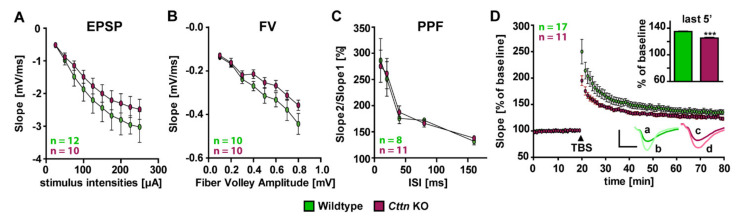
**Electrophysiological characterization of *Cttn*-knockout mice.** (**A**,**B**) Characterization of basal synaptic transmission via measurements of excitatory postsynaptic potentials (EPSP) as well as input-output measurements (‘Fiber Volley’, FV) in acute hippocampal slices from WT and Cttn-deficient animals. (**C**) Analysis of short-term plasticity in WT and *Cttn*-KO acute hippocampal slices as evaluated by paired-pulse facilitation (PPF). (**D**) Long-term potentiation induced via theta-burst stimulation (TBS) after 20 min of baseline recording at the Schaffer collateral to CA1 pathway in acute hippocampal slices derived from WT and *Cttn*-KO mice. Representative single fEPSP responses are shown for (a) WT Ctrl (b) WT LTP (c) *Cttn* KO Ctrl (d) *Cttn* KO LTP. Scale Y = 1 mV, X = 5 ms. **Data information:** n = number of slices; Statistic tests used: For normal distribution of data: Kolmogorov-Smirnov Test, Parametric Tests: Two-Way ANOVA and Sidak’s Multiple Comparisons, Non-Parametric Tests: Kruskal-Wallis Test and Dunn’s test or Student’s T-Test; data depicted as means ± SEM, detailed information on *p* values as well as statistical tests is depicted in Appendix A; significances are indicated by *** *p* value < 0.001.

**Figure 3 cells-10-01835-f003:**
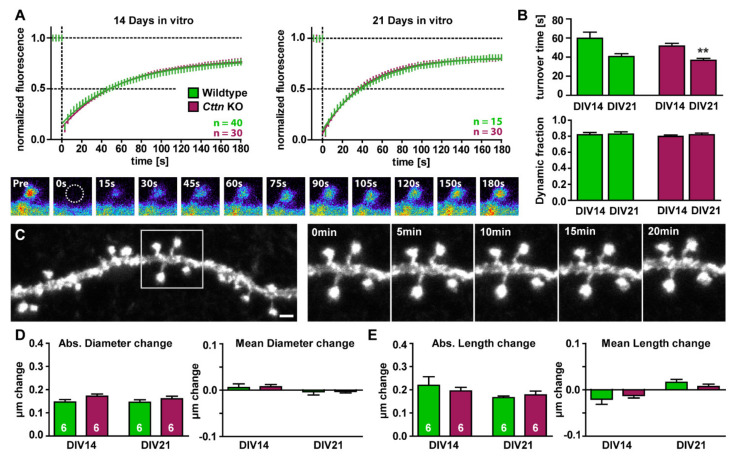
**Basal actin dynamics and dendritic spine motility are normal in Cttn-deficient hippocampal neurons.** (**A**) Basal actin dynamics as analyzed via fluorescence recovery after photobleaching (FRAP) of EGFP-ß-actin in primary dissociated hippocampal neurons after 14 as well as 21 days in culture. Depicted are the recovery curves of hippocampal WT and *Cttn*-KO spines under baseline conditions. Representative images show fluorescence levels derived from EGFP-ß-actin of an individual dendritic spine before bleaching as well as during recovery of fluorescence over a time course of 180 s after the bleaching pulse. n = number of dendritic spines analyzed. (**B**) Quantification of the turnover time and the fraction of dynamic actin (the amount of actin fluorescence which recovered in the time period of 180 s after the photobleach). (**C**) Analysis of dendritic spine motility. Representative images depict an individual dendritic stretch that was repeatedly imaged every 5 min for a total of 20 min. Scale bar = 1 µm. (**D**,**E**) Absolute as well as mean changes in (**D**) the head diameter and the (**E**) length of individual dendritic spines in 14- or 21-day-old primary dissociated hippocampal cultures from WT and *Cttn*-KO mice. n = number of dendrites analyzed. **Data information:** Statistic tests used: For normal distribution of data: Kolmogorov-Smirnov Test, Parametric Tests: Two-Way ANOVA and Sidak’s Multiple Comparisons, Non-Parametric Tests: Kruskal-Wallis Test and Dunn’s test; data depicted as means ± SEM, detailed information on *p* values as well as statistical tests is depicted in Appendix A; significances are indicated by ** *p* value < 0.01.

**Figure 4 cells-10-01835-f004:**
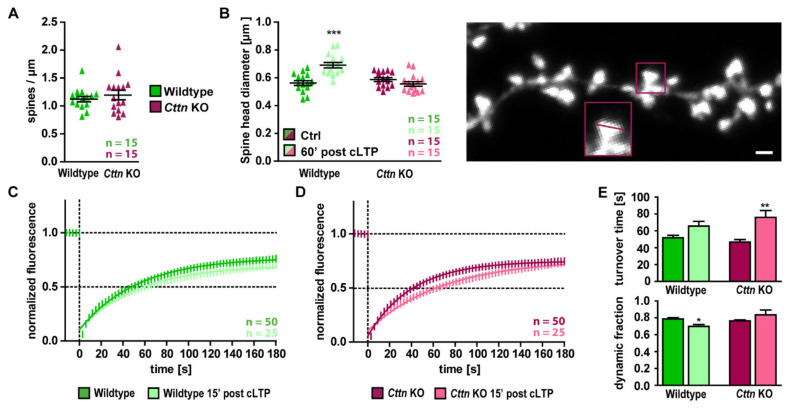
**Structural plasticity is completely absent in Cttn-deficient hippocampal neurons.** (**A**) Analysis of dendritic spine density of hippocampal WT and *Cttn*-KO neurons. n = number of dendrites analyzed. (**B**) Analysis of the average dendritic spine head diameter under basal conditions as well as 60 min after induction of chemically induced NMDAR-mediated long-term potentiation (cLTP). Note that structural plasticity (i.e., growth of dendritic spine head diameter) is completely missing in Cttn-deficient hippocampal neurons. Representative image depicts maximum intensity projection image of a healthy individual dendritic segment and an example of a dendritic spine head diameter analysis. n = number of dendrites analyzed. Scale bar = 1 µm. (**C**,**D**) Recovery curves (EGFP-ß-actin) of individual dendritic spines from (C) WT and (D) *Cttn*-KO hippocampal neurons under baseline conditions as well as in the first 15 min after chemical induction of LTP. (**E**) Quantification of the turnover time and the fraction of dynamic actin (the amount of actin the fluorescence of which recovered in the time period of 180 s after the photobleach) under baseline conditions as well as in the first 15 min after chemical induction of LTP. n = number of dendritic spines analyzed. **Data information:** Statistic tests used: For normal distribution of data: Kolmogorov-Smirnov Test, Parametric Tests: Two-Way ANOVA and Sidak’s Multiple Comparisons, Non-Parametric Tests: Kruskal-Wallis Test and Dunn’s test; data depicted as means ± SEM, detailed information on *p* values as well as statistical tests is depicted in Appendix A; significances are indicated by * *p* value < 0.05, ** *p* value < 0.01 and *** *p* value < 0.001.

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
