# Peer review of "Cortactin Contributes to Activity-Dependent Modulation of Spine Actin Dynamics and Spatial Memory Formation"

_cells, 2021, doi:10.3390/cells10071835_

Round 1

Reviewer 1 Report

In the manuscript entitled “Cortactin contributes to activity-dependent modulation of spine actin dynamics and spatial memory formation”, Cornelius et al. used several experimental assays to demonstrate that cortactin is important to spatial memory formation through regulation of functional and structural spine plasticity via actin polymerization. Overall, I believe the manuscript is very well written and presents solid results. However, a revision is required before being accepted for publication. My comments are listed below.

1) Figure 4B: the authors quantified the spine head diameters for WT and Cttn KO, both in control and 60' post cLTP conditions. They found a significant increase in spine head diameter for WT after 60' post Cltp (plot). However, the fluorescence microscopy images do not exactly show this variation. I understand that the variation is quite small (0.55 and 0.69µm). Thus, I suggest the authors should remove the images. If they want, they could insert an image of a representative dendritic spine at high magnification to indicate how the measurement (spine head diameter) is performed; however, this image should come before the plot.

2) Figures 4C, 4D and 4E: The authors quantified what they called turnover time and dynamic fraction using FRAP. The turnover time is the halftime of recovery? The dynamic fraction is the mobile fraction? If so, the authors should try to find better representative curves for the conditions.

Author Response

In the manuscript entitled “Cortactin contributes to activity-dependent modulation of spine actin dynamics and spatial memory formation”, Cornelius et al. used several experimental assays to demonstrate that cortactin is important to spatial memory formation through regulation of functional and structural spine plasticity via actin polymerization. Overall, I believe the manuscript is very well written and presents solid results. However, a revision is required before being accepted for publication. My comments are listed below.

  • Figure 4B: the authors quantified the spine head diameters for WT and Cttn KO, both in control and 60' post cLTP conditions. They found a significant increase in spine head diameter for WT after 60' post Cltp (plot). However, the fluorescence microscopy images do not exactly show this variation. I understand that the variation is quite small (0.55 and 0.69µm). Thus, I suggest the authors should remove the images. If they want, they could insert an image of a representative dendritic spine at high magnification to indicate how the measurement (spine head diameter) is performed; however, this image should come before the plot.

We agree that the increase in the dendritic spine head diameter after cLTP was not properly visible in the presented images. However, we originally included them not only as representative images for the analysis but also in order to depict that the analyzed neurons are generally healthy - therefore, we decided against removing them completely. Nonetheless, we thankfully took the advice and now only show one image of a healthy dendrite at higher magnification and indicate how the dendritic spine head diameter analysis is performed. This image is now presented together with the dendritic spine head diameter analysis plot in Figure 4B.

Reviewer 2 Report

Cornelius et al. looked at neuronal function of Cortactin by using a Cortactin KO mouse model. They characterized several basic parameters (such as behavioral analysis and e-phys), and looked at the actin dynamics (in basal conditions and upon chemical LTP stimulation). Authors concluded that Cortactin KO mice have a memory deficit, a LTP deficit, and that in Cortactin KO neurons chemical LTP-induced increase in spine head diameter is abolished. They proposed that these phenotypes are caused by changes in spine actin dynamics observed in Cortactin KO neurons, which in my opinion is the less strong part of the paper. Overall, the manuscript describes interesting phenotypes, but does not provide mechanistic insight on the Cortactin function at the spine compared to previous work (for example, Hering and Sheng 2003; Tanaka et al., 2020 and Alicea et al., 2017). In my opinion authors need to address several points:

- Authors claimed that the phenotypes described in their ms are “explained by a reduction of activity-dependent modulation of actin polymerization in cortactin-deficient neurons”, lines 24-25. In my opinion this an overstatement. Authors followed actin dynamics (during basal conditions and upon chemical LTP induction) with FRAP, but they did not look at actin polymerization. Authors need to tune down their conclusion or to perform experiments to validate this statement. It would be nice if authors would follow actin polymerization in basal conditions and upon chemical LTP induction in both genotypes (for instance see 10.1021/bc100595z, Ishimoto et al., 2011).

- In Fig. 4C-E authors concluded that upon cLTP induction the turnover time of F-actin is increased in Cortactin KO neurons compared to WT neurons. In my opinion, authors here need to compare WT (cLTP) with KO (cTLP). Are these two groups different? Importantly, what is this data (rather small differences in F-actin dynamics) telling us about the general Cortactin function, and in particular about Cortactin function in actin polymerization? This is not clear in the current version of the manuscript. In this regard, I think authors should also discuss their findings in light of a recent publication (10.1093/jmicro/dfaa001, Tanaka et al., 2020). Here, they described that actin turnover was enhanced in dendritic spines of Cortactin KO neurons.

- In my opinion the data presented does not support strongly an effect on activity-dependent modulation of actin dynamics (see previous comment), therefore I would remove this idea from the title.

 - As well introduced by the authors, previous work (Hering and Sheng 2003) showed that Cortactin loss of function decreased the density of spines. However, here authors could not replicate this finding (Fig. 4A). Could authors please discuss this discrepancy?

- The protocol of chemical LTP used by the authors contains small variations compared to the protocol frequently used by the group of Robert Malenka (for instance Ahmad et al., 2012), which has been validated and used by different groups. Namely, the protocol here used does not contain TTX and Picrotoxin in the extracellular solution. Was this protocol used and validated it somewhere else? If not, did the authors validate this protocol (e-phys and/or synaptic AMPARs immunostaining)?

- Statistical tests and statistical information are provided in a supplementary table. In my opinion this information should be added to Fig. legends, which will help the reader to interpret the data. Related to this point, did authors check for normal distribution? It looks like only parametric tests were used. Do all data groups follow a normal distribution? Which statistical test did authors use for this analysis? This information needs to be added. Finally, what are their statistical units (mice, slices, neurons, etc) throughout the different Figures? This information is missing in Fig. legends.

- Total distance traveled is indicated as data not shown, I would add this to Fig. 1, next to panel A.

- Differences in LTP (Fig. 2D) are not very convincing. Can authors also show amplitude of EPSP? Do they find differences here? What was the statistical test used to compare the groups? What do they use as a statistical unit (mice or slices)? This should be added to the Fig. legend.

Author Response

Cornelius et al. looked at neuronal function of Cortactin by using a Cortactin KO mouse model. They characterized several basic parameters (such as behavioral analysis and e-phys), and looked at the actin dynamics (in basal conditions and upon chemical LTP stimulation). Authors concluded that Cortactin KO mice have a memory deficit, a LTP deficit, and that in Cortactin KO neurons chemical LTP-induced increase in spine head diameter is abolished. They proposed that these phenotypes are caused by changes in spine actin dynamics observed in Cortactin KO neurons, which in my opinion is the less strong part of the paper. Overall, the manuscript describes interesting phenotypes, but does not provide mechanistic insight on the Cortactin function at the spine compared to previous work (for example, Hering and Sheng 2003; Tanaka et al., 2020 and Alicea et al., 2017). In my opinion authors need to address several points:

  • Authors claimed that the phenotypes described in their ms are “explained by a reduction of activity-dependent modulation of actin polymerization in cortactin-deficient neurons”, lines 24-25. In my opinion this an overstatement. Authors followed actin dynamics (during basal conditions and upon chemical LTP induction) with FRAP, but they did not look at actin polymerization. Authors need to tune down their conclusion or to perform experiments to validate this statement. It would be nice if authors would follow actin polymerization in basal conditions and upon chemical LTP induction in both genotypes (for instance see 10.1021/bc100595z, Ishimoto et al., 2011).

We agree that none of our experiments allow a conclusion on the effect that the loss of Cttn could have on actin polymerization directly and we apologize for the unintended overstatement in our abstract. To correct for that, we now changed the sentence to ‘These phenotypes might at least in part be explained by alterations in the activity-dependent modulation of synaptic actin in cortactin-deficient neurons.’

  • In Fig. 4C-E authors concluded that upon cLTP induction the turnover time of F-actin is increased in Cortactin KO neurons compared to WT neurons. In my opinion, authors here need to compare WT (cLTP) with KO (cTLP). Are these two groups different? Importantly, what is this data (rather small differences in F-actin dynamics) telling us about the general Cortactin function, and in particular about Cortactin function in actin polymerization? This is not clear in the current version of the manuscript. In this regard, I think authors should also discuss their findings in light of a recent publication (10.1093/jmicro/dfaa001, Tanaka et al., 2020). Here, they described that actin turnover was enhanced in dendritic spines of Cortactin KO neurons.

We indeed included the comparison of cLTP in both genotypes in our statistical analysis. Neither the turnover time nor the dynamic actin fraction is statistically different when comparing WT cLTP vs KO cLTP, respectively, therefore also no significances were depicted in Figure 4E. However, we believe that it is especially important to look at alterations mediated upon induction of cLTP within each genotype. Here it becomes obvious that WT spines behave differently than KO spines following induction of synaptic plasticity. Hippocampal dendritic spines from Cttn KO animals show a statistically significant increase in the turnover time following cLTP induction when compared to ctrl spines which WT spines do not, while on the other hand, WT spines show a statistically significant decrease of the dynamic actin fraction in this early phase following cLTP that is not visible in Cttn KO spines. We agree that these changes are rather small, however, they suggest that activity-dependent modulations of synaptic actin might actually differ on the molecular level in Cttn KO animals, suggesting a context-dependent regulatory function of Cttn.

Our findings presented here, in light of the recent publication from Tanaka and colleagues (2020), are largely supporting their observations. In line with our data, they did not find changes in dendritic spine density, spine morphology as well as spine dynamics in neurons from conventional Cttn KO mice (deletion of exon 5). In contrast to our data they saw enhanced actin turnover under basal conditions as indicated by a significant difference in the time constant calculated from the equation describing the fluorescence recovery over time in FRAP experiments. However, they took individual images only every 15s for up to 120s to follow fluorescence recovery (per spine 8 images post-bleach in total) which on the one hand, does not allow for a precise depiction of the fluorescence recovery over time given the low amount of images (as indicated by their rather large error bars) and on the other hand, is too short of a time period to analyze fluorescence recovery. It is clearly visible that their recovery curves did not reach the plateau phase yet and, in our opinion, they do not convincingly show that actin turnover times are indeed different between the genotypes. Given the fact that their FRAP curves under basal conditions are nearly overlapping, similar to what our data suggests in Figure 3A,B, we believe that actin turnover in the Cttn KO under basal conditions is rather unaltered.

Nevertheless, we agree that additional experiments will be needed to gain more insight, especially, since stated correctly, our data does not offer any information on the underlying molecular mechanisms or on the effect that the loss of Cttn could have on actin polymerization directly. While future studies will need to focus on analyzing the direct role of Cttn in actin polymerization and activity-dependent modifications of synaptic actin as well as the molecular pathways involved, we believe this would extend the scope of our work presented here.

  • In my opinion the data presented does not support strongly an effect on activity-dependent modulation of actin dynamics (see previous comment), therefore I would remove this idea from the title.

It is true that the loss of Cttn shows a rather mild phenotype with regard to dendritic spine actin dynamics (which might be in parts due to compensational mechanisms in conventional KO mice also discussed in answer #4). Yet, our data suggests that the modulation of actin dynamics in response to cLTP is different in Cttn KO neurons when compared to WT neurons. We indeed did not want to overemphasize our findings and therefore already chose a wording that we still believe is appropriate with regard to all phenotypes described in our paper (ranging from behavior to electrophysiology and spine plasticity) that Cttn ‘contributes’ to activity-dependent regulations of synaptic actin, we believe that it is justified to keep the idea in the title.

  • As well introduced by the authors, previous work (Hering and Sheng 2003) showed that Cortactin loss of function decreased the density of spines. However, here authors could not replicate this finding (Fig. 4A). Could authors please discuss this discrepancy?

Indeed Hering and Sheng (2003) were able to show that the acute loss of Cortactin had a very profound impact on the dendritic spine density in hippocampal neurons. More precisely, they used an acute siRNA-mediated knockdown of Cttn in hippocampal cultures at DIV13 and analyzed those 72 h later at DIV16 where they showed that the dendritic spine density drops from 0.377 spines per µm to 0.126 spines per µm after the loss of Cttn. In our experience acute loss of a protein might lead to stronger alterations in cultured cells compared to conventional KO mice. We observed a similar but even more pronounced discrepancy for the loss of profilin2a (acute knock down resulted in a reduction in spine density, Michaelsen 20210, conventional KO showed no alteration in synapse density, Pilo Boyl 2007). While a conventional Cttn KO mouse model (deletion of exon 7) has the advantage to study behavior and electrophysiology in the absence of cortactin compensatory mechanisms might indeed mask effects that would rather be visible in an acute knockdown approach. Supporting this, in a study of Tanaka et al. (2020) (which was also already mentioned in reviewer question #2) which analyzed a conventional Cttn KO mouse model quite similar to ours (carrying a deletion of Cttn exon 5), changes in the dendritic spine density of hippocampal neurons in 3 week but also 10 week old animals were not detected - which is line with our observations.

  • The protocol of chemical LTP used by the authors contains small variations compared to the protocol frequently used by the group of Robert Malenka (for instance Ahmad et al., 2012), which has been validated and used by different groups. Namely, the protocol here used does not contain TTX and Picrotoxin in the extracellular solution. Was this protocol used and validated it somewhere else? If not, did the authors validate this protocol (e-phys and/or synaptic AMPARs immunostaining)?

Glycine-mediated LTP has long been known to exhibit characteristic features of stimulus-induced LTP (Shahi and Baudry, 1993) and the protocol without TTX and Picrotoxin has already been validated elsewhere e.g. by Fortin et al., 2010. They could show that glycine-LTP induced by this protocol increases the dendritic spine head area, the mEPSC amplitude, the synaptic expression of AMPARs as well as surface levels of GluA1. In addition, they could confirm that glycine-mediated LTP requires NMDAR signaling, CAMKI activation, actin polymerization as well as AMPAR-mediated activation of the Rac-PAK-LIMK pathway. We also showed reliable LTP following Glycine (without TTX and picrotoxin) application in cultured hippocampal neurons (Michaelsen-Preusse et al. 2016 supplementary Figure 2C).

  • Statistical tests and statistical information are provided in a supplementary table. In my opinion this information should be added to Fig. legends, which will help the reader to interpret the data. Related to this point, did authors check for normal distribution? It looks like only parametric tests were used. Do all data groups follow a normal distribution? Which statistical test did authors use for this analysis? This information needs to be added. Finally, what are their statistical units (mice, slices, neurons, etc) throughout the different Figures? This information is missing in Fig. legends.

We added information on the statistical units throughout the different figures as indeed, this information was missing in the legends. In addition, we tested for normal distribution utilizing the Kolmogorov-Smirnov test. Only in two cases (see below) we re-analyzed data utilizing the Kruskal-Wallis Test followed by a post-hoc Dunn’s test with selected pairs of columns as the data were not normally distributed. We added the ‘new’ statistical information into the Supplementary Table S1. Overall, we had to include two important changes:

  1. Figure 1A (behaviour in the Open Field), where for both genotypes we had to remove the significance between the percentage of time spent in the center and the middle region of the arena. However, as animals of both genotypes behaved almost identical in the Open Field this did not change the interpretation of the results.
  2. Figure 3B (FRAP in DIV14 vs DIV21 cultures) where both genotypes show a decrease of the turnover time in DIV21 cultures when compared to DIV14 cultures. Now, the decrease is statistically significant in the Cttn KO, but not in the WT. However, as still, both genotypes behave very similar in this respect this does not alter the interpretation of our data.

Finally, we agree that statistical information is important to interpret the results and should be easily assessable to the reader. Yet, we believe that the summary of statistical information in one compact table (S1) has the advantage that everything can be assessed easily in one place and moreover that the reading of text and figure legends is not disturbed. As a compromise, we added the tests used and the p values thresholds for significances depicted in the figures into the figure legends while we kept the complete set of statistical information in Supplementary Table S1.

  • Total distance traveled is indicated as data not shown, I would add this to Fig. 1, next to panel A.

We now added the values describing the total distance travelled in the Open Field experiment in the correct paragraph of the results section and removed the ‘data not shown’.

  • Differences in LTP (Fig. 2D) are not very convincing. Can authors also show amplitude of EPSP? Do they find differences here? What was the statistical test used to compare the groups? What do they use as a statistical unit (mice or slices)? This should be added to the Fig. legend.

While it is difficult to assess which degree of difference in the end is biologically meaningful, we do believe that the range of data presented in our story offers sufficient supporting evidence that alterations in hippocampal LTP in Cttn KO animals indeed could have a biological impact on the behavioral level. In addition to the two LTP curves depicted in Fig.2D (WT vs KO), which are clearly distinct from each other over the whole recording period following Theta burst stimulation, we now added representative single-fEPSP responses as an insert into the same Figure and included descriptive information into the Figure legend. Note that fEPSP size and shape was normal and that in addition, the fEPSP amplitude is plotted against varying stimulus intensities as well as Fiber volley amplitude in Figure 2A and 2B, respectively.

As already mentioned in reviewer question #6, information on the statistical units was added into the Figure legends, while detailed information on the statistical tests used is summarized in Supplementary Table S1.

Reviewer 3 Report

In this study, Cornelius et al., investigated the role of Cortactin in synaptic plasticity and learning and memory using the Cortactin KO mouse. Although basal synaptic structure and function seems normal in these animals, the authors discovered that synaptic plasticity in the form of LTP – both functional and structural- was impaired in these animals. Consistent with these cellular deficits, learning and memory was also impaired.

Overall, this study is consistent with the notion that actin and actin-binding proteins are essential for synaptic plasticity and memory. However, it fell short of providing a mechanistic insight on how cortactin might be regulating synaptic plasticity- though this was probably out of the scope of this study.

I only have minor concerns and as such I recommend its publication once addressed.

1.- In general, the LTP and learning and memory deficits found in the cortactin KO are only modest and as such suggest that cortactin plays only modulatory roles rather in LTP and memory. Although this is clear from this manuscript figures, this is not reflected in the text where it is written for instance that “knockout of Cttn leads to a robust deficit in hippocampus-dependent spatial memory formation” in line 225. For robust deficits in the morris water maze, please refer to the PSD95 KO (Migaud et al., 1998) or CaMKII KOs (Silva et al., 1992). I would recommend tuning down the “robust “ effect.

2.- Although the Cdc42 and Rac KOs also display deficits in LTP and memory, the are not “strikingly similar” to the cortactin KO as argued in the discussion (line 428). For instance, the deficits in LTP in both the cd42 and rac ko are much more pronounced than those found in the cortactin KO. Again, I would recommend tuning down these parallels, especially if not experimental evidence is provided.

Author Response

In this study, Cornelius et al., investigated the role of Cortactin in synaptic plasticity and learning and memory using the Cortactin KO mouse. Although basal synaptic structure and function seems normal in these animals, the authors discovered that synaptic plasticity in the form of LTP – both functional and structural- was impaired in these animals. Consistent with these cellular deficits, learning and memory was also impaired.

Overall, this study is consistent with the notion that actin and actin-binding proteins are essential for synaptic plasticity and memory. However, it fell short of providing a mechanistic insight on how cortactin might be regulating synaptic plasticity- though this was probably out of the scope of this study.

I only have minor concerns and as such I recommend its publication once addressed.

  • In general, the LTP and learning and memory deficits found in the cortactin KO are only modest and as such suggest that cortactin plays only modulatory roles rather in LTP and memory. Although this is clear from this manuscript figures, this is not reflected in the text where it is written for instance that “knockout of Cttn leads to a robust deficit in hippocampus-dependent spatial memory formation”in line 225. For robust deficits in the morris water maze, please refer to the PSD95 KO (Migaud et al., 1998) or CaMKII KOs (Silva et al., 1992). I would recommend tuning down the “robust “ effect.

We took the advice and made sure not to overemphasize the effect in our manuscript. Therefore, we reworked the mentioned section in our results part and now speak of ‘a moderate deficit’ rather than a robust one.

  • Although the Cdc42 and Rac KOs also display deficits in LTP and memory, the are not “strikingly similar” to the cortactin KO as argued in the discussion (line 428). For instance, the deficits in LTP in both the cd42 and rac ko are much more pronounced than those found in the cortactin KO. Again, I would recommend tuning down these parallels, especially if not experimental evidence is provided.

As suggested, we tuned down the wording used to describe the parallels in behavioral and molecular phenotypes between Cdc42, Rac1 and Cttn KO animals. To avoid a semantic reinforcement of the phenotypic similarities between these animal models by using the term ‘strikingly’, we now changed it to ‘comparable’.

Round 2

Reviewer 1 Report

The authors have answered all comments raised by this reviewer

Reviewer 2 Report

Authors have improved their ms considerable, and I support its publication now.